# Instance Segmentation in Very High Resolution Remote Sensing Imagery Based on Hard-to-Segment Instance Learning and Boundary Shape Analysis

**Yiping Gong** [1] , **Fan Zhang** [1] , **Xiangyang Jia** [1] , **Zhu Mao** [1] , **Xianfeng Huang** [1,2,]* **and Deren Li** [1]

1   The State Key Laboratory of Information Engineering in Surveying, Mapping and Remote Sensing, Wuhan University, 129 Luoyu Road, Wuhan 430079, China; gongyp15@whu.edu.cn (Y.G.); zhangfan@whu.edu.cn (F.Z.); jiaxiangyang@whu.edu.cn (X.J.); maoz@whu.edu.cn (Z.M.); drli@whu.edu.cn (D.L.)
2   Institute of Yangtze River Civilization and Archaeology, Wuhan University, 129 Luoyu Road, Wuhan 430079, China
*   Correspondence: huangxf@whu.edu.cn

**Abstract:** Although great success has been achieved in instance segmentation, accurate segmentation of instances remains difficult, especially at object edges. This problem is more prominent for instance segmentation in remote sensing imagery due to the diverse scales, variable illumination, smaller objects, and complex backgrounds. We find that most current instance segmentation networks do not consider the segmentation difficulty of different instances and different regions within the instance. In this paper, we study this problem and propose an ensemble method to segment instances from remote sensing images, considering the enhancement of hard-to-segment instances and instance edges. First, we apply a pixel-level Dice metric that reliably describes the segmentation quality of each instance to achieve online hard instance learning. Instances with low Dice values are studied with emphasis. Second, we generate a penalty map based on the analysis of boundary shapes to not only enhance the edges of objects but also discriminatively strengthen the edges of different shapes. That is, different areas of an object, such as internal areas, flat edges, and sharp edges, are distinguished and discriminatively weighed. Finally, the hard-to-segment instance learning and the shape-penalty map are integrated for precise instance segmentation. To evaluate the effectiveness and generalization ability of the proposed method, we train with the classic instance segmentation network Mask R-CNN and conduct experiments on two different types of remote sensing datasets: the iSAID-Reduce100 and the JKGW_WHU datasets, which have extremely different feature distributions and spatial resolutions. The comprehensive experimental results show that the proposed method improved the segmentation results by 2.78% and 1.77% in mask AP on the iSAID-Reduce100 and JKGW_WHU datasets, respectively. We also test other state-of-the-art (SOTA) methods that focus on inaccurate edges. Experiments demonstrate that our method outperforms these methods.

**Keywords:** instance segmentation; hard-to-segment instance learning; boundary shapes analysis; remote sensing imagery

## 1. Introduction

Alongside advances in deep convolutional neural networks, a series of state-of-the-art tasks including classification [1–4], object detection [5–10], semantic segmentation [11–13] and instance segmentation [14–18] have been proposed. As a high-level task that can yield both correct detection and precise segmentation of an object, instance segmentation has received extensive attention and has become a fundamental and meaningful technique for many visual applications such as mapping, environmental management, and urban planning and monitoring. Although great achievements have been made, instance segmentation still faces the problem of poorly segmented objects, especially at object edges, and requires more delicate network design (see Figure 1).

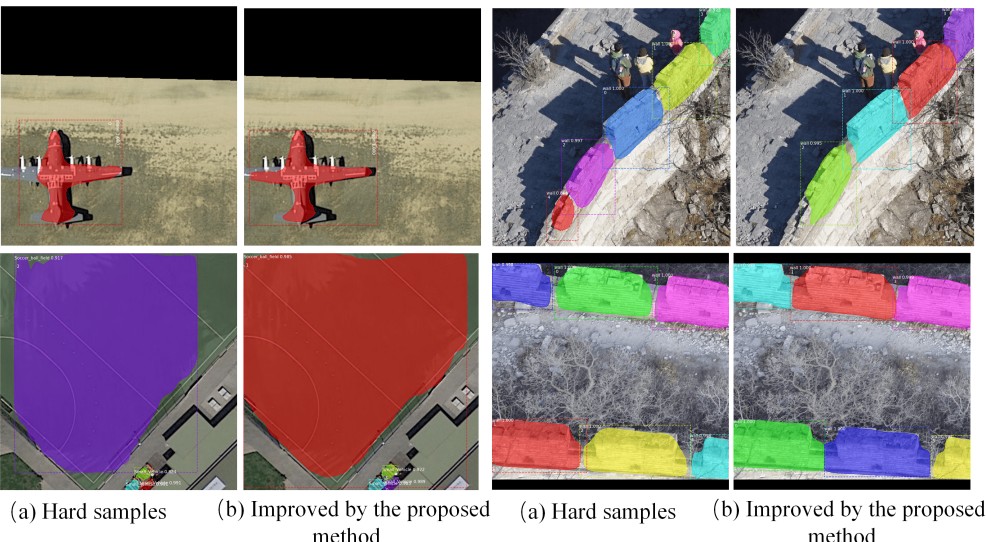

(a) Hard samples     (b) Improved by the proposed method     (a) Hard samples     (b) Improved by the proposed method

**Figure 1.** We propose an ensemble method to tackle the problem of inaccurate instance segmentation in remote sensing imagery, which takes advantage of hard instance learning and boundary shape analysis. (**a**) depicts hard samples that commonly exist and hamper the instance segmentation. (**b**) shows the improved segmentation results using the proposed method.

As shown in Figure 1, the main challenge of the instance segmentation task is the inaccurate segmented objects. Compared with internal areas, the edges of objects are often poorly segmented. Due to the spatial resolution, changes in lighting, complex backgrounds, etc., edge pixels may contain chaotic information mixed with the background, making them more difficult to distinguish from background pixels. In other word, the edge pixels are more susceptible to the background environment. This problem is more prominent for pixels near sharp edges. However, most advanced instance segmentation networks are basically designed for large objects in natural images. A large object size makes the object easy to capture, and a clear boundary between the object and the background makes the segmentation accuracy very high. Therefore, these networks, such as Mask R-CNN [15], MaskLab [19] and Mask Scoring R-CNN [16], generally do not pay special attention to poorly segmented objects and object edges. When these networks are directly used for instance segmentation in very-high-resolution (VHR) remote sensing imagery, in which object sizes are smaller and object boundaries are more complicated, the segmentation accuracy decreases. Consequently, it is still challenging to segment instances from remote sensing imagery. To tackle this problem, we propose an ensemble method to pay more attention to hard instances and regions of an instance during the training process. We demonstrate that the commonly used instance segmentation network lacks the ability to strengthen hard-to-segment instances and their edges and can be substantially improved by better network design towards the aforementioned issues.

Unlike these works, we focus on the enhancement of poorly segmented instances and their edges. To achieve this goal, our method estimates the segmentation difficulty of each instance and assigns discriminative weights to different parts of the object.

In terms of segmentation difficulty, inspired by the hard example learning method with Focal Loss [20] and the intersection-over-union (IoU) metric of semantic segmentation that reliably describes the instance segmentation quality, we propose integrating a pixel-level Dice indicator that is smoother than the intersection-of-union (IoU) into the network to represent the segmentation difficulty of each instance. Instances with low Dice values will be learned with emphasis online. For the enhancement of object edges, many state-of-the-art methods have been proposed [21–26]. These methods fall into two categories: one involves refining the coarse mask prediction with edges, and the other directly upweights edge pixels using edge information such as gradient, morphological gradient and distance information. The former method introduces a subtask for edge detection, which leads

to increased computational cost, while the latter method does not make any changes to the network structure and is simpler and more convenient. Nevertheless, neither of the methods address the issue that sharp edges are more likely to be misclassified than flat edges. Therefore, in consideration of the boundary shapes, we propose a shape-penalty method that gives more attention to pixels located at sharp edges than flat edges. Finally, the methods of hard instance learning and boundary shape-penalty are combined for instance segmentation.

To summarize, we propose an ensemble method to tackle the problem of inaccurate instance segmentation in remote sensing imagery, which takes advantage of hard instance learning and boundary shape analysis. The major contributions of our work are the following three folds:

1.  We propose an ensemble workflow to strengthen the learning of hard-to-segment instances and edges, especially sharp edges, and explore our method on two datasets with different feature distributions and spatial resolutions.
2.  For hard instance mining, we adopt a pixel-level Dice metric that reliably describes the segmentation quality of each instance to achieve online hard instance learning. To the best of our knowledge, it is the first attempt to calculate and apply the segmentation difficulty of instances for hard instance learning.
3.  Edges of different shapes are explored and weighted by the shape-penalty map discriminatively, which effectively improves the performance of instance segmentation detectors.

The remainder of this paper is organized as follows. In Section 2, related work, including instance segmentation, hard example mining and edge enhancement, is reviewed. In Section 3, the proposed framework is carefully illustrated. Experiments and analysis are presented in Section 4. The discussion and conclusion are given in Sections 5 and 6, respectively.

## 2. Related Work

### 2.1. Instance Segmentation

Instance segmentation requires both the correct detection and precise segmentation of an object. Therefore, current methods can be roughly categorized into two classes: detection-based methods and segmentation-based methods.

Driven by the great success of region-based CNNs such as R-CNN [5], Fast R-CNN [6], Faster R-CNN [7] and numerous extensions [8–10], detection-based methods begin with object recognition and then generate object segmentation masks by filtering out background pixels within each object region. Pinheiro et al. [27] proposed DeepMask to segment and classify the center object in a sliding window fashion. Dai et al., 2016 [14], presented multitask network cascades for instance-aware semantic segmentation, which consists of three networks that respectively differentiate instances, estimate masks, and categorize objects. He et al. [15] proposed Mask R-CNN, which is built in top of Faster R-CNN by introducing a branch for predicting an object mask. Chen et al. [19] proposed MaskLab, which uses position-sensitive scores to obtain better results. Huang et al. [16] extended the mask branch in Mask R-CNN with a MaskIOU block, which uses the instance feature and the corresponding predicted mask together to regress the mask IoU. An underlying problem in these methods is that the segmentation accuracy is affected by the detection accuracy.

Segmentation-based methods are inspired by the development of semantic segmentation methods such as the fully convolutional network (FCN) [11], U-Net [12], and DeepLab [13]. This type of method begins with pixel-level segmentation and then groups pixels to form different instances according to their object classes. Dai et al., 2016 [17] proposed instance-sensitive FCNs to generate position-sensitive maps and then assembled them to obtain instance-level masks. Li et al. [28] used position-sensitive maps with inside/outside scores to generate instance segmentation results. Anurag et al. [18] used category-level segmentation, along with cues from the output of an object detector, within an end-to-end CRF to predict instances. Alexander et al. [29] proposed InstanceCut, which cuts semantic segmentation results into different instances by detecting instance



boundaries. Other works, such as [30,31], generated region-level segmentation proposals using multiscale combinatorial grouping [30] and then extracted the features of each segmentation proposal before performing classification.

Neither of the above classes of methods considers the enhancement of hard instances and hard edges. When objects are under complex conditions, such as objects with small sizes, indefinite boundaries and changing illumination intensities, the segmentation difficulty increases. In this case, the segmentation accuracy decreases.

### 2.2. Hard Example Mining

Hard example mining is a widely studied issue in the field of object detection and has shown effectiveness in accuracy improvement [32]. However, in the instance segmentation task, no work is performed to extract hard-to-segment instances. Therefore, we focus on this problem and propose our method. There are usually two types of methods that focus on the extraction and enhancement of hard objects: data-based methods and loss-based methods.

Data-based methods select hard examples to update the dataset for iterative training. In training support vector machines (SVMs), hard examples are those that violate the current model's margin and are used to replace easy examples that are correctly classified beyond the current model's margin [33]. When this rule is applied, only a small subset of the entire training set is used to implement the SVM classifier. When training shallow neural networks or boosted decision trees [34], false positives are seen as hard examples and are added to the original dataset for training again. However, this training usually involves only one iteration and is not suitable for models trained online. To address this problem, Abhinav et al. [9] proposed an online hard example mining (OHEM) algorithm that pushes all proposals forward to obtain loss values and trains the model based only on the selected hard RoIs. Since only some of the data are involved in the training, the time consumption is reduced. However, the selection of hard examples generates additional computations.

In contrast, loss-based methods emphasize hard examples by assigning different weights to easy and hard examples. In these methods, all examples are used for training, but their contributions during the learning process are different. Focal Loss [20] reshapes the standard cross-entropy loss to downweight easy examples and thus focuses training on hard negatives. Since no additional modules are added, this type of method can save memory consumption. Inspired by this work, we propose a method that focuses the model on hard instances by assigning different weights to different instances. Hard instances will receive higher weights, while easy instances will receive lower weights. To calculate the instance weight, we use the pixel-level classification results to estimate the instance segmentation difficulty.

### 2.3. Edge Enhancement

Inaccurate segmentation at object edges has long been a problem limiting the accuracy of instance segmentation. To improve the segmentation accuracy at object edges, many methods have been proposed. Some methods, such as Gated-SCNN [21] and that proposed by Liang et al. [22], add a branch for edge detection in parallel with the existing branch for object mask prediction. The detected edge information is used to refine the coarse mask prediction. The difference between the two methods is that Gated-SCNN proposes a spatial attention module that uses edge prediction to strengthen the coarse segmentation result, while the second method applies a domain transform to well align the object segmentation with the object boundaries. BMask R-CNN [35] extends the Mask R-CNN by explicitly utilizing object boundary information to improve mask-level positioning accuracy. However, these methods bring about increased computational cost due to the extra edge detection task.

Other methods indicate that edge pixels should be more emphasized than other pixels during training; therefore, they use edge information such as gradient, morphological gradient and distance information to assign higher weights to edge pixels. Contour Loss [25] regards edge pixels as hard-to-segment pixels and upweights edge pixels with morphological gradients derived from ground-truth masks. Francesco et al. [26] weighted all pixels inside the object using a distance mask in which pixels in proximity to edges are weighted more than those located far from edges. Boundaryloss [23] was proposed as a novel surface loss aimed at minimizing the distance between the ground-truth boundary and the prediction boundary. These methods give larger weight values to pixels close to object boundaries than those far from boundaries. Nevertheless, these methods do not address the issue that pixels at sharp edges are more likely to be misclassified than those at flat edges. Therefore, in this work, we propose a method that not only enhances edge pixels but also distinguishes pixels located at different edges to give more attention to pixels located at sharp edges than flat edges.

## 3. The Method

### 3.1. Motivation

Due to problems of complicated backgrounds, occlusion, illumination intensity variations, object sizes, etc., instances are difficult to segment well, and instance edges, especially sharp edges, are easily misclassified. While in most current region-based instance segmentation networks, the instance mask is obtained by performing a simple foreground/background classification within the object proposal, without considering hard instances and hard edges. Consequently, it is still challenging to segment instances from remote sensing imagery. To tackle this problem, we propose an ensemble method to give more attention to hard instances and hard regions of an instance during the training process.

We use the widely used instance segmentation network Mask R-CNN as the baseline and embed our method into it, as shown in Figure 2 (on the left). There are two main stages of our approach. First, to focus the model more on hard instances, the segmentation difficulty of each instance is calculated based on the predicted mask and ground-truth mask. The instance segmentation difficulty reliably depicts the quality of segmentation, as well as the difficulty of segmentation, which decays to one as the IoU of the predicted mask and ground-truth mask increases. Then, a shape-penalty map is generated from the label image through the analysis of boundary shapes, in which pixels at sharp edges are weighted more than those at flat edges and in internal areas of the object. Since the shape-penalty map is generated offline, it doesn't result in an increase in training time. Finally, the segmentation difficulty and the shape-penalty map are integrated for final instance segmentation. A comparison of results obtained with and without our method is also provided in Figure 2 (on the right). The details of each aspect are described in the following sections.

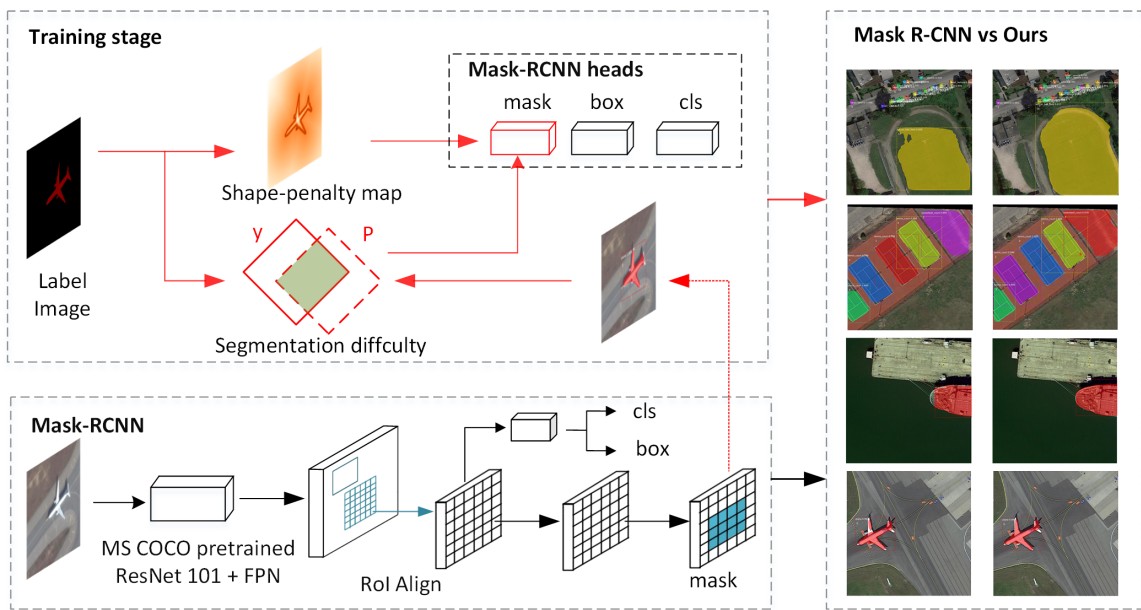

**Figure 2.** The overall workflow of the proposed method built in Mask R-CNN. There are two main stages of our approach. First, the instance segmentation difficulty that reliably describes the instance segmentation quality is calculated for emphasising hard instances. Then, a shape-penalty map is generated offline to discriminatively strengthen hard edges of different shapes, in which pixels at sharp edges are weighted higher than those at flat edges and in internal areas of the object. Finally, the instance segmentation difficulty and the shape-penalty map are combined and used at the mask head for precise mask prediction. The segmentation results obtained with and without our method are shown on the right.

### 3.2. Hard-to-Segment Instance Learning

Instances with low segmentation quality are seen as hard instances. For segmentation quality estimation, the commonly used criteria are the pixel-level IoU and the Dice coefficient. Some works [36,37] directly use the Dice coefficient or the IoU as the objective function, pursuing precise segmentation results by minimize the difference between the predicted mask and its ground-truth mask. However, when using IoU or Dice loss functions for training, the training gradient usually changes dramatically, leading to erratic training. Thus, the training curves may not be credible. Unlike these works, we use IoU or Dice criteria to represent the instance segmentation difficulty and reweight all instances based on these criteria.

We choose the Dice coefficient, which is a smoother indicator than the IoU, to depict the instance segmentation difficulty. The Dice coefficient is inversely related to the difficulty of segmentation. Thus, the segmentation difficulty of each instance is written as the inverse of the Dice coefficient:

$$Dice = \frac{2 * \mid y \cap P \mid}{|y| + |P|} = \frac{2 * TP}{FN + FP + 2 * TP}$$

$$ins\_w = \frac{1}{Dice} * |P - y| + (1 - |P - y|)$$

$$(1)$$

*Dice* measures the overlap degree of *P* and *y*, where *P* is the predicted mask derived from the probability map with a threshold of 0.5 and *y* is the ground-truth mask. *ins_w* is the instance weight that gradually approaches 1 as the prediction accuracy improves. Instead of penalizing all pixels of the instance, the factor $\frac{1}{Dice}$ is applied only to misclassified pixels, including false foregrounds (or false positives, FP) and false backgrounds (or false negatives, FN), which achieves the best results in our experiments. For correctly classified regions, the weight value equals 1. Figure 3 illustrates *y*, *P*, *TP*, *TN*, *FP* and *FN*.

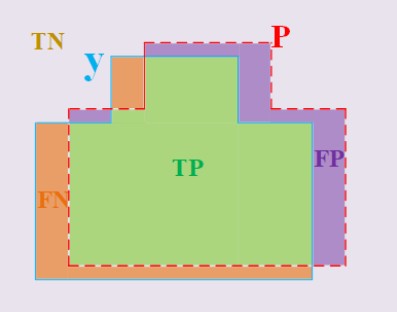

**Figure 3.** Illustration of *y*, *P*, *TP*, *TN*, *FP* and *FN*.

### 3.3. Shape-Penalty Map for Edge Enhancement

The segmentation difficulty differs not only between instances but also between regions within a single instance. Intuitively, edges pixels are surrounded by more background pixels, compared with the center pixels of an instance. Coupled with a small instance size, changes in lighting, etc., edge pixels may contain chaotic information mixed with the background, making them more difficult to distinguish from background pixels. In other word, the edge pixels are more susceptible to the background environment. This problem is more prominent for pixels near sharp edges. To enhance the learning of edge pixels, especially sharp edges, we propose a shape-penalty method based on the analysis of the category complexity around the pixels, which shows the intuitive difference between non-edge pixels and edge pixels, as well as edges of different shapes. Through the generated shape-penalty map, different areas of the instance, such as internal areas, sharp edges and flat edges are assigned different weights.

From the aspect of categories, internal pixels are mostly surrounded by pixels of the same category, while edge pixels are generally surrounded by pixels of different categories. Obviously, pixels located at sharp edges are usually surrounded by more pixels of different categories than those located at flat edges and in internal regions. Therefore, the shape-penalty method can distinguish not only edges and interiors, as in [23,25,26], but also different edges. Similar to edge-detection algorithms, we define a local neighborhood area of $7 \times 7$ pixels to generate category information for each pixel. The shape-penalty map is defined as:

$$
\begin{aligned}
num\_sc &= \mathbb{C}_{7\times7} \odot y \\
cls\_ratio &= \frac{num\_dc}{num\_sc} = \frac{\Omega_{7\times7} - num\_sc}{num\_sc} \\
ratio\_norm &= \frac{cls\_ratio}{\max(cls\_ratio)} + 1 \\
edge\_w &= \begin{cases} ratio\_norm & cls\_ratio > 0 \\ f(distance, \delta) & cls\_ratio = 0 \end{cases} \\
\delta &= \min(ratio\_norm) \\
edge\_w &= \frac{edge\_w}{\max(edge\_w)}
\end{aligned}
\tag{2}
$$

as shown in Equation (2), a convolutional operation $\odot$ is first performed on the ground-truth map *y* to count the number of pixels in the same category as the center pixel. Note that the foreground (red region, plane area) and the background (black region) are counted separately. When dealing with the plane area, the red region is marked as 1 and the black region is marked as 0. While dealing with the background area, the black region is marked as 1 and the red region is marked as 0. Then, the ratio of different categories (num_dc) to the same category (num_sc) is generated to describe the category complexity around the pixel, where $\Omega_{7\times7}$ is a metric that statistics the total number of pixels in the current local area. The *cls_ratio* value is further normalized to 0~1, and to avoid a zero weight, an additional value of 1 is added. For edge pixels with *cls_ratio* larger than 0, *ratio_norm* is

directly used as the weight, while for internal pixels with *cls_ratio* equal to 0, a function $f$ is applied to attenuate the weight from the area furthest from the center of the instance. The process can be reduced to two steps. First, we calculate the pixelwise distance for each non-edge pixel by applying the method proposed in [26] and normalize the distance to 0~1. Then, we use the minimum value of the edge pixel weight $\delta$ as the initial attenuation value and multiply it with the pixelwise distance. In this way, the weight of non-edge pixels decreases gradually from $\delta$ to 0 as they approach the center of the object. Finally, all weight values are normalized to 0~1. Figure 4 shows the ground truth, $7 \times 7$ kernel, and shape-penalty map in detail. In contrast to [26], the proposed method distinguishes edges with the same pixelwise distance to the instance center based on the analysis of their local category distribution, thus giving more attention to those sharp edges.

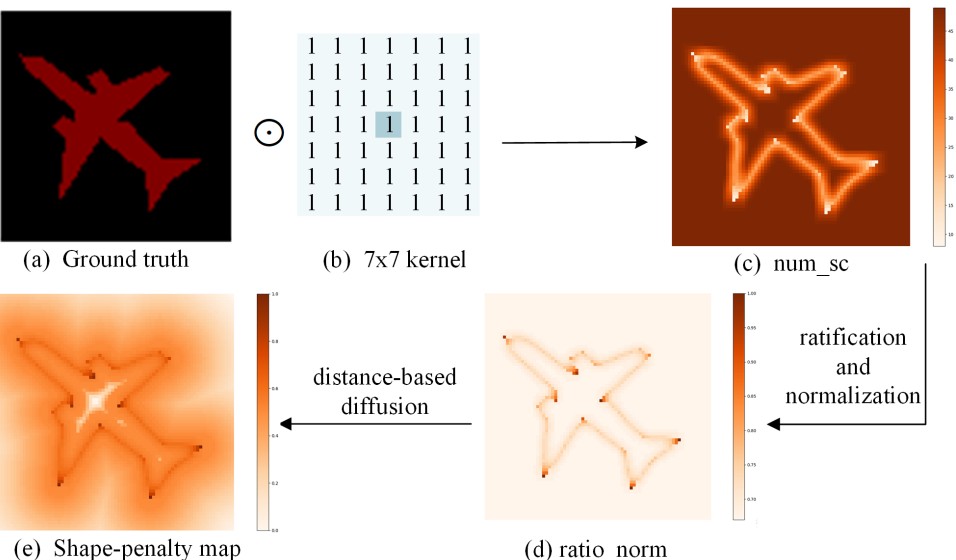

(a) Ground truth  (b) 7x7 kernel  (c) num_sc

ratification and normalization

distance-based diffusion

(e) Shape-penalty map  (d) ratio_norm

**Figure 4.** Shape-penalty map. (**a**) Ground truth label. (**b**) Convolution with a $7 \times 7$ kernel. (**c**) Number of pixels with the same category as the center. Note that the plane area (red region) and the background area (black region) are counted separately. (**d**) Ratio of different categories (num_dc) to the same categories (num_sc). (**e**) Shape-penalty map, in which the weights at sharp edges are larger than those at flat edges and in internal areas of an object.

### 3.4. Mask R-CNN Improved by Our Method

3.4.1. Architecture

Mask R-CNN is a classic region-based instance segmentation network that can be simply regarded as Fast R-CNN and mask prediction. Fast R-CNN is made up of classification and bounding box regression. The mask branch is a small FCN applied to each RoI, which predicts a segmentation mask in a pixel-to-pixel manner. Since the category of an RoI is preclassified in the object-detection branch, the mask branch performs only a simple foreground/background classification, which helps avoid competition between different classes. However, Mask R-CNN doesn't take into account poorly segmented objects and object edges.

To improve the performance of Mask R-CNN on hard instances and hard edges, we embed our method into Mask R-CNN. First, the shape-penalty map of every training example is generated offline from the label image and then fed to the network along with the training examples. Next, the instance segmentation difficulty is calculated online at the mask head using the prediction and the ground truth. Finally, the shape-penalty map and the segmentation difficulty are combined to construct a weight matrix to act on the BCE loss function in the mask branch for precise mask prediction. The final mask prediction loss integrated with our approach is shown in Equation (3), where $N$ is the number of pixels, $y_i$ is the ground-truth label of the $i$th pixel, and $p_i$ is the predicted probability given by the

classifier at the end of the network. Parameters $ins\_w_i$ and $edge\_w_i$ are the instance-level and pixel-level weight factors, respectively.

$$
\begin{aligned}
Loss_{our}(y, p) = &-\frac{1}{N}\sum_{i=1}^{N} W_i * (y_i * log(p_i) + \\
&(1 - y_i) * log(1 - p_i)) \\
W_i = &ins\_w_i * edge\_w_i
\end{aligned}
\tag{3}
$$

### 3.4.2. Training and Inference

In the training stage, training examples are sent to the network to obtain the prediction results. Then, the loss is calculated based on the predicted results and the ground truth and is backpropagated to the front of the network to optimize the parameters layer by layer. To reduce the computational cost, Mask R-CNN resizes the outputs of the network and ground truth masks to a fixed size of (28,28) and provides an upsampling module to achieve segmentation of the input image sizes. We follow the network parameters, such as the RPN anchors, the number of sampled RoIs in each image, and the ratio of positive to negatives, provided in [15]. The network is trained using the stochastic gradient descent (SGD) algorithm.

The inference stage is a simple forward process that passes an image through the network and obtains the results, including the instance category, instance bounding box and instance mask. During testing, the number of proposals for the FPN is 1000, and then, only the 100 highest-scoring detection boxes among these proposals are fed to the mask branch to generate instance masks [15]. The $28 \times 28$ floating-number mask output is then resized to the RoI size and binarized at a threshold of 0.5. This is the standard Mask R-CNN inference procedure. We also follow this procedure.

## 4. Experiments and Results

Our experiments use Mask R-CNN with a ResNet-101 + FPN backbone, which is a classic region-based instance segmentation network, as the baseline. In this section, we first show a thorough comparison of the proposed method to the state of the art on two datasets with different pixel resolutions: the iSAID-Reduce100 and the JKGW_WHU datasets (Section 4.3). Then, we provide an in-depth analysis on the iSAID-Reduce100 dataset by conducting comprehensive ablations to validate our architecture design (Section 4.4). The experimental results are displayed using the standard COCO evaluation metrics, including bounding box average precision (bbox AP) and mask average precision (mask AP), to evaluate object detection and instance segmentation, respectively. Bbox AP and mask AP are both averaged over an IoU of 0.5.

### 4.1. Datasets

#### 4.1.1. iSAID-Reduce100

The iSAID [38] dataset is the largest remote sensing dataset for instance segmentation, containing a total of 2806 remote sensing images. The images have various spatial resolutions, which are collected from the Google Earth, satellite JL-1 and satellite GF-2 of the China Centre for Resources Satellite Data and Application. Since the image sizes range from ~800 × 800 pixels to ~4000× 13,000 pixels, which are too large for training with Mask R-CNN, we crop the images to 512 × 512 pixels using a sliding window spanning 410 pixels. However, for a region-based instance segmentation network, training all cropped images on one GPU is very time-consuming; thus, we create a reduced version of iSAID, which is called iSAID-Reduce100, as shown in Figure 5. The iSAID-Reduce100 dataset is composed of 100 training images and ~100 testing images in each category. We use the predefined training set to train the models and the validation set to evaluate the models.

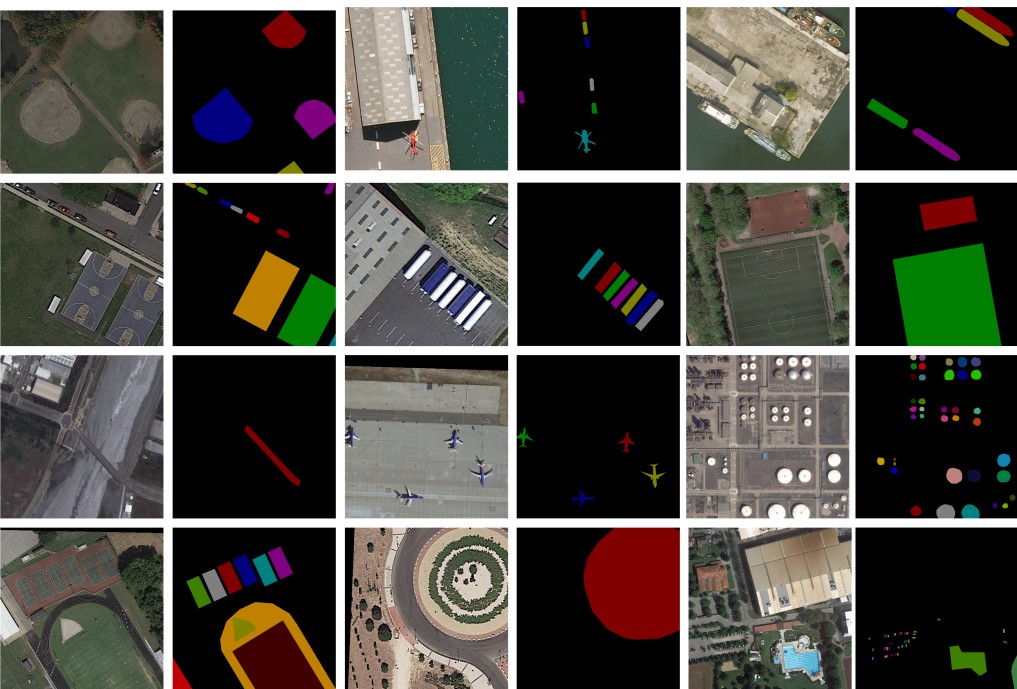

**Figure 5.** Illustration of the iSAID-Reduce100 dataset. There may be different objects in an image.

### 4.1.2. JKGW_WHU

The dataset of the Jiankou Great Wall is captured by a Falcon 8+ drone, containing a total of 625 UAV images with a 4912 × 7360 pixel resolution. The ground sample distance is approximately 0.5 cm to 3 cm. We randomly select 334 images for training, 90 images for validation and 191 images for testing. Since the pixel resolution of the UAV images is too large for the CNN to process, we crop the image to 1024 × 1024 pixels. Cropped images that contain Great Wall merlons are selected for the experiments, including 2773 training images, 404 validation images and 1621 testing images. The ground truth is annotated manually via the open annotation tool LabelMe, as shown in Figure 6.

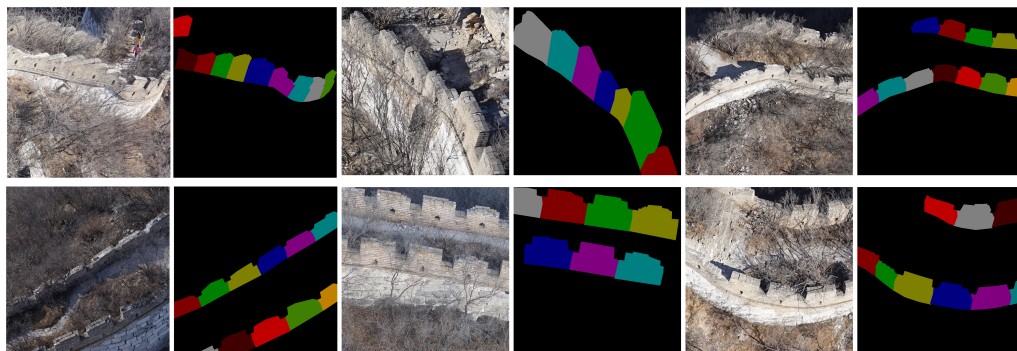

**Figure 6.** Illustration of the JKGW_WHU dataset.

### 4.2. Experimental Setup

We build the proposed approach on the successful Mask R-CNN, which is pretrained on the MS COCO dataset. The code is implemented as in [39]. For the iSAID-Reduce100 dataset, whose image size is 512 pixels, the number of images processed by each GPU is 4. For the JKGW_WHU dataset with an image size of 1024 pixels, the number of images processed by each GPU is 1, due to memory limitations. For both datasets, we train on one GPU (Nvidia TITAN XP, 12 GB of memory) for 30,000 iterations, with a learning rate of 0.0001. We use a weight decay of 0.0001 and a momentum of 0.9.

*4.3. Experimental Results*

We compare our method to state-of-the-art methods in instance segmentation, including binary cross entropy (BCE) Loss, Focal Loss, ContourLoss and Distance-Penalty Loss, in Tables 1 and 2. BCE Loss is the original loss function of Mask R-CNN. Focal Loss is designed to tackle the problem of imbalance between foreground and background classes in one-stage object detection networks. In two-stage networks such as Mask R-CNN, these loss functions filter out most background examples in the RPN; thus, we abandon the factor $\alpha$ and use only $(1 - pt)^2$ to weight each pixel in the mask subnet. ContourLoss assigns the same weight value to all edge pixels empirically. Distance-Penalty Loss weights pixels using a distance map, where pixels near the edge are weighted higher than those far from the edge. We implement the above methods on the Mask R-CNN mask subnet and test their performance according to the bbox AP and mask AP.

4.3.1. Instance Segmentation on iSAID-Reduce100

As shown in Table 1, the proposed method achieves the best segmentation results, reaching 42.57% bbox AP and 41.54% mask AP. The relevant works listed in Table 1 show improvement both in bbox AP and mask AP, except Focal Loss. Directly using Focal Loss (only $(1 - pt)^2$) to penalize pixels does not seem to work in instance segmentation. Concerning mask AP, compared with the way of strengthening all the edges with no discrimination, our method further promotes edge enhancement by discriminatively weighting edges of different shapes, outperforming ContourLoss and Distance-Penalty Loss. In addition, we evaluate the performance of the hard-to-segment instance learning method and the shape-penalty map separately, as shown in Table 1. The results show the effectiveness of the proposed two strategies on both bbox AP and mask AP. At last, compared with the original Mask R-CNN with the standard BCE Loss, our method improves bbox AP and mask AP by 1.05% and 2.78%, respectively. The Mask R-CNN, ContourLoss, Distance-Penalty Loss (the second-best performer) and the proposed method outputs on the iSAID-Reduce100 val set are visualized in Figure 7.

**Table 1.** Instance segmentation comparison of BCE Loss, Focal Loss, ContourLoss, Distance-Penalty Loss and the proposed method on the iSAID-Reduce100 val set.

| Method | Backbone | bbox AP (%) | Mask AP (%) |
|---|---|---|---|
| BCE Loss [15] | ResNet-101 | 41.52 | 38.76 |
| Focal Loss [20] | ResNet-101 | 40.83 | 37.80 |
| ContourLoss [25] | ResNet-101 | 41.76 | 40.19 |
| Distance-Penalty Loss [26] | ResNet-101 | 41.48 | 41.02 |
| Hard-to-segment instance learning | ResNet-101 | 41.98 | 40.55 |
| Shape-penalty map | ResNet-101 | 42.36 | 41.02 |
| The proposed method | ResNet-101 | 42.57 | 41.54 |

However, there are still two issues. (1) Although the proposed method helps detect and segment objects, some objects can not be well detected, such as small vehicles obscured by trees, large vehicles with black texture (see the fourth and fifth rows of Figure 7), etc. To address this issue, better object detectors are required. (2) Our approach can not completely solve the problem of inaccurate edge segmentation (see the second row of Figure 7) and more efforts should be made.

4.3.2. Instance Segmentation on JKGW_WHU

Due to the single object category and the large object size, the segmentation accuracy of oblique images is higher than that of aerial images. As shown in Table 2, compared with BCE, the proposed method brings 1.67% and 1.77% performance gains in bbox AP and mask AP, respectively. Compared to Focal Loss, ContourLoss and distance-penalty loss, which also focus on the learning of hard pixels or edge regions, our method obtains the best results, demonstrating the superiority of our strategies of strengthening hard instances and regions. The segmentation results obtained by using the hard-to-segment instance learning method and

the shape-penalty map are also provided in Table 2. Figure 8 shows the segmentation results of Mask R-CNN trained with the BCE Loss, ContourLoss, Distance-Penalty and our method. As can be seen, for objects with an extreme tilting perspective, our method achieves the best results among them, especially at object edges.

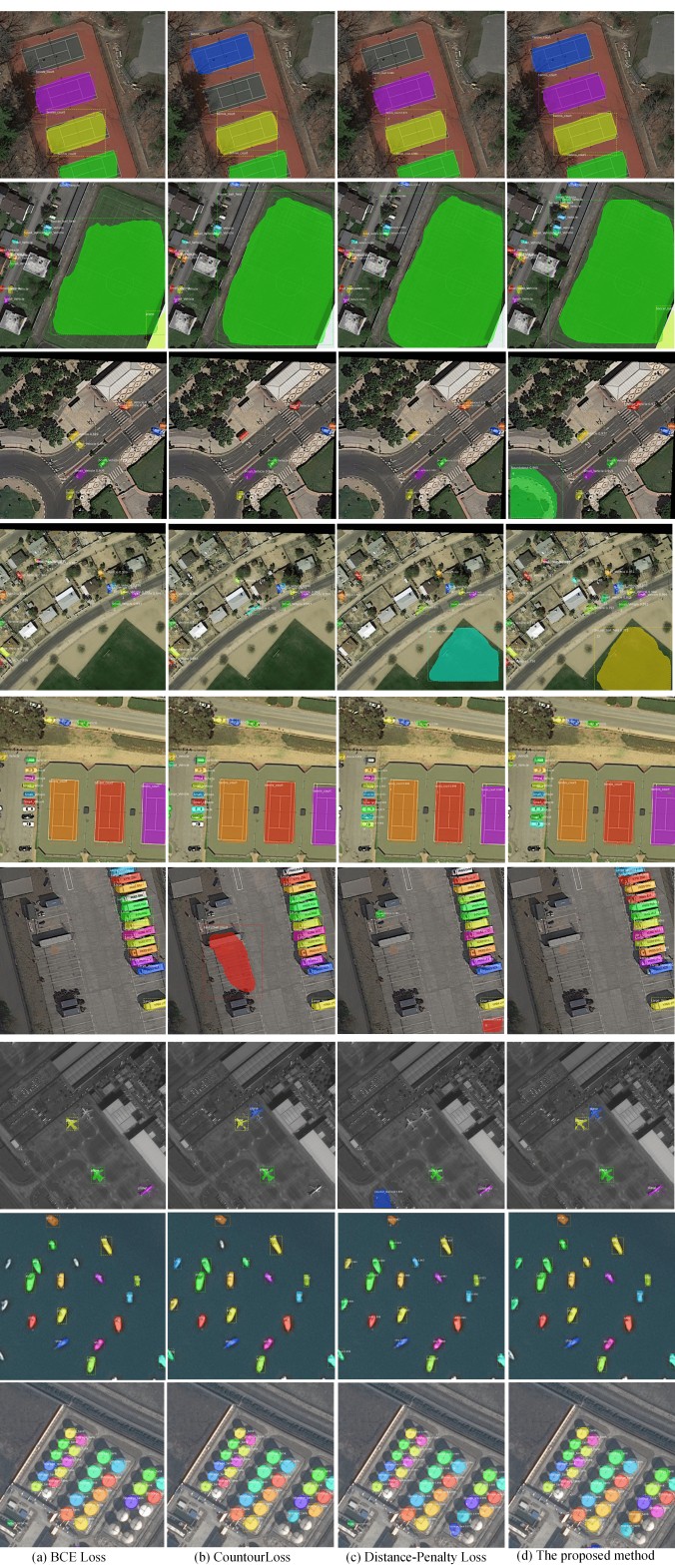

**Figure 7.** Instance segmentation results of Mask R-CNN trained with BCE Loss, ContourLoss, Distance-Penalty Loss and the proposed method.

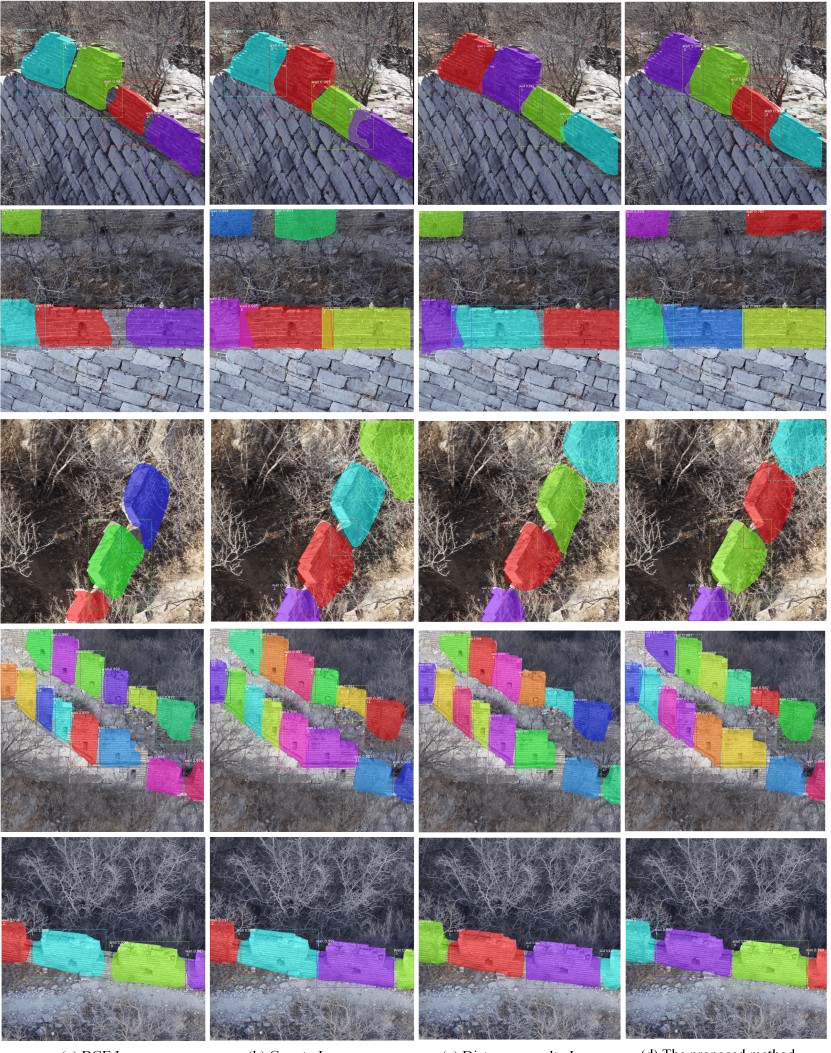

(a) BCE Loss  (b) CounterLoss  (c) Distance-penalty Loss  (d) The proposed method

**Figure 8.** Instance segmentation results of Mask R-CNN trained with BCE Loss, ContourLoss, Distance-Penalty Loss and the proposed method.

**Table 2.** Instance segmentation comparison of BCE Loss, Focal Loss, ContourLoss, Distance-Penalty Loss and the proposed method on the JKGW_WHU test set.

| Method | Backbone | bbox AP (%) | Mask AP (%) |
|---|---|---|---|
| BCE [15] | ResNet-101 | 92.43 | 92.35 |
| Focal Loss [20] | ResNet-101 | 92.47 | 92.54 |
| ContourLoss [25] | ResNet-101 | 93.17 | 93.15 |
| Distance-penalty Loss [26] | ResNet-101 | 93.17 | 93.36 |
| Hard-to-segment instance learning | ResNet-101 | 93.90 | 93.85 |
| Shape-penalty map | ResNet-101 | 92.51 | 93.45 |
| The proposed method | ResNet-101 | 94.10 | 94.12 |

### 4.3.3. Quantitative Comparison with Advanced Methods

Furthermore, we evaluate the effectiveness of the proposed method by comparing with other advanced methods, including Mask Scoring R-CNN and BMask R-CNN. The quantitative results on the two datasets are shown in Table 3. To make this a fair comparison, we train all the methods with the pretrained models on the ImageNet dataset for 120,000 iterations. Obviously, our method that explicitly reinforces hard objects and edges performs better than these models. We also notice that the performance varies with different pretrained models. This is why the increase of bbox AP is smaller than before.

Compared with the original Mask R-CNN, we improve mask AP on the two datasets by 3.18% and 2.57%, respectively.

**Table 3.** Quantitative comparison of Mask R-CNN, Mask Scoring R-CNN, BMask R-CNN and the proposed method on the iSAID-Reduce100 val set and the JKGW_WHU test set.

| Method | Backbone | iSAID-Reduce100 | | JKGW_WHU | |
| | | bbox AP (%) | Mask AP (%) | bbox AP (%) | Mask AP (%) |
| --- | --- | --- | --- | --- | --- |
| Mask R-CNN [15] | ResNet-101 | 41.47 | 35.24 | 92.97 | 93.20 |
| Mask Scoring R-CNN [16] | ResNet-101 | 41.27 | 36.14 | 92.17 | 92.30 |
| BMask R-CNN [35] | ResNet-101 | 39.02 | 36.46 | 91.23 | 91.32 |
| The proposed method | ResNet-101 | 41.52 | 38.42 | 93.64 | 95.77 |

### 4.4. Architecture Design Analysis

In this section, we provide in-depth analysis of the architecture, hard-to-segment instance learning, edge enhancement based on the shape-penalty map and timing on the iSAID-Reduce100 val set. We use ResNet-101 FPN for the last two ablation experiments. Since the improvement in bbox AP is small, we mainly discuss the performance on the mask AP. Their effects on instance segmentation are discussed in detail next.

#### 4.4.1. Architecture

We report our results on different backbone networks in Table 4 to verify the robustness and portability of our approach. In our experiments, the proposed method significantly improves the segmentation results on both the ResNet-50 and ResNet-101 backbones. When using shallow networks as the backbone, such as ResNet-50, our method achieves a greater improvement, improving the segmentation results by 1.35% in bbox AP and 6.99% in mask AP. Additionally, ResNet-101 achieves better results than ResNet-50, indicating that deeper networks are helpful in instance segmentation. We note that not all frameworks automatically benefit from deeper or advanced networks [40].

**Table 4.** Results of BCE Loss and the proposed method using ResNet-50 FPN and ResNet-101 FPN as backbones on the iSAID-Reduce100 val set.

| Method | Backbone | bbox AP | Mask AP |
| --- | --- | --- | --- |
| BCE Loss | ResNet-50 | 39.36% | 32.57% |
| The proposed method | ResNet-50 | 40.71% | 39.56% |
| BCE Loss | ResNet-101 | 41.52% | 38.76% |
| The proposed method | ResNet-101 | 42.57% | 41.54% |

#### 4.4.2. Hard-to-Segment Instance Learning

To understand the hard-to-segment instance learning method better, we analyze different methods of penalizing an instance, including weighting all pixels (ALL) or weighting only false pixels (FP and FN). Figure 9 shows the results obtained using our hard-to-segment instance learning. We can see that weighting only false pixels is better than weighting all pixels, improving the segmentation results by 1.01% in mask AP. This is because the former method further reduces the learning of simple areas of the instance, thereby forcing the model more toward areas that are not well-learned. It is similar to the hard example mining strategy in the field of object detection [9,20], which reduces the learning of simple objects. Finally, compared with the classic Mask R-CNN, weighting only false pixels obtains a 1.79% improvement in mask AP.

#### 4.4.3. Edge Enhancement Based on the Shape-Penalty Map

The shape-penalty map is used to identify and discriminatively enhance the edges of different shapes, which leverages local category information obtained through a convolution operation. In theory, the larger the size of the convolution kernel, the more obvious the

boundary features are. However, is it true that the larger the convolution kernel, the better the shape description is? To answer this question, we explore the influence of different sizes of convolution kernels on instance segmentation.

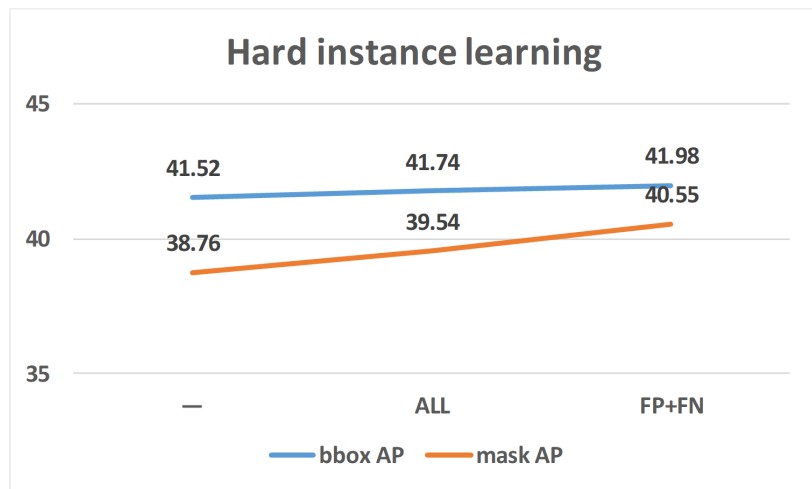

**Figure 9.** Effects of hard-to-segment instance learning. "−" indicates the results of Mask R-CNN without using the hard-to-segment instance learning method.

Figure 10 shows the ablation results of edge enhancement with different convolution kernel sizes, including $3 \times 3$, $5 \times 5$, $7 \times 7$ and $9 \times 9$. As shown, as the kernel size increases, the segmentation accuracy gradually increases at first. However, when the kernel size exceeds $7 \times 7$, for example, $9 \times 9$, the segmentation accuracy decreases. This is because the boundary shape feature is a low-level feature. Therefore, if the convolution kernel is too large, it may contains too much information related to other edges, which is less relevant to the current central pixel and interferes with the extraction of shape. Moreover, a too large window can also cause the weight of the edge to be much greater than the weight of the internal region, resulting in extremely unbalanced learning of different parts of the instance. This is a bit of an overcorrection. In our experiments, the $7 \times 7$ convolution kernel achieves the best results, which improves the segmentation accuracy by 2.26%. We regard the $7 \times 7$ convolution kernel as approximately optimal. In the application, $7 \times 7$ can be used as the starting size to search for the best kernel size.

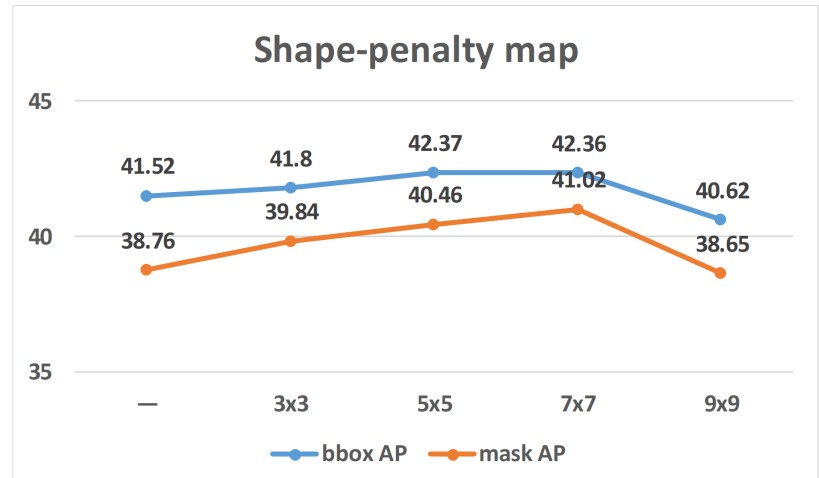

**Figure 10.** Effects of the shape-penalty map generated using different kernel sizes. "−" indicates the results of Mask R-CNN without using the shape-penalty map.

### 4.4.4. Timing

Training with ResNet-50-FPN on the iSAID-Reduce100 val set takes 14 h in our 1-GPU implementation and 22 h with ResNet-101-FPN. Compared with the classic Mask R-CNN, the training time was increased by less than one hour since the calculation of segmentation difficulty and the reading of the offline calculated shape-penalty map. In addition, the shape-penalty map brings about additional memory usage when fed into the network along with the training images and ground truth masks.

In the inference stage, this model runs at 312 ms per image with a size of $512 \times 512$ pixels on an Nvidia TITAN XP GPU. Since there is no change in the architecture, our approach does not increase the architecture parameters and inference time.

## 5. Discussion

We propose an ensemble method for accurate instance segmentation in VHR remote sensing imagery with the benefit of segmentation difficulty description and boundary shape analysis. The proposed method consists of two stages. First, to focus the model more on hard instances, the segmentation difficulty of each instance is calculated based on the predicted mask and ground-truth mask. Then, to put more attention to not only hard instances but also hard regions within each instance, a shape-penalty map is generated from the label image through the analysis of boundary shapes, in which pixels at sharp edges are weighted higher than those at flat edges and in internal areas of the object. Finally, the two methods are integrated for precise instance segmentation.

However, there are still three limitations. (1) First, the proposed method is specifically designed for better segmentation of objects. Although the experiments show that the proposed method helps detect and segment the objects from the background, it is still hard to address very tiny objects with occlusion, less texture and bad imaging conditions. (2) Second, concerning the extraction of boundary shape information, we define a local region and use the category distribution within the region to determine the edge shape. The appropriate region size is selected by testing different window sizes, which lacks theoretical guidance. Therefore, the selected window size is just approximately optimal. (3) Finally, the shape-penalty maps generated offline require additional storage space and will cause an increase in memory usage when fed to the network along with images and label images.

## 6. Conclusions

In this paper, we concentrate on the two challenges that always exist in the instance segmentation task: poorly segmented instances and instance edges, especially sharp edges. To solve these problems, we propose an ensemble method, which generates the segmentation difficulty of each instance to enhance the learning of hard-to-segment instances and utilizes the local category distribution to distinguish and discriminatively weigh edges of different shapes. The proposed method was tested on two datasets with different feature distributions and spatial resolutions: the iSAID-Reduce100 and the JKGW_WHU datasets. The comprehensive experimental results on the iSAID-Reduce100 dataset show that we improve bbox and mask AP by 1.05% and 2.78%, respectively. On the JKGW_WHU dataset, our method brings 1.67% and 1.77% performance gains in bbox AP and mask AP, respectively. Although the performance of the segmentation is improved by considering the hard objects and edges, the shortcomings still exist. We regard our work as the starting point to explore hard-to-segment instance learning and enhancement of edges of different shapes. We hope that our work will be useful for subsequent work.

**Author Contributions:** Conceptualization, F.Z., X.H. and D.L.; Data curation, X.H.; Formal analysis, Y.G., F.Z., X.J., Z.M., X.H. and D.L.; Funding acquisition, F.Z.; Methodology, Y.G.; Writing – original draft, Y.G.; Writing—review & editing, Y.G., F.Z., Z.M., X.H. and D.L. All authors have read and agreed to the published version of the manuscript.

**Funding:** This research was funded by the National Key R&D Program of China (No.2020YFC1523003) and the National Key R&D Program of Hubei province (No.2020BAB125).

**Institutional Review Board Statement:** Not applicable.

**Informed Consent Statement:** Not applicable.

**Data Availability Statement:** Code and datasets have been made available at: https://github.com/primegong/Instance-Segmentation-in-VHR-Imagery (accessed on 10 November 2021).

**Conflicts of Interest:** The authors declare no conflict of interest.

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
