# Peer review of "Instance Segmentation in Very High Resolution Remote Sensing Imagery Based on Hard-to-Segment Instance Learning and Boundary Shape Analysis"

_remotesensing, doi:10.3390/rs14010023_

Round 1

Reviewer 1 Report

The article proposes a new loss function based on the problem of hard example segmentation in instance segmentation. The author uses instance weight and shape-penalty map to construct a weight matrix to act on the cross-entropy loss function. By optimizing the cross-entropy loss function in instance segmentation, the problem of hard example segmentation is improved. My question is:

  1. The numdc in formula 2 in the second paragraph of 3.3 in the article needs to be consistent with the num_dc in the third paragraph. Can 7*7-num_sc in Equation 2 be clearly described as subtracting pixel by pixel, because cls_ratio is a ratio metric?
  2. There is a question here. In fig3, fig(b) is used to convolve fig(a). I think the result can’t be fig(c). The value of the plane area in fig(c) is correct in my opinion, but the value of the background area should be 0, which the color is light yellow.
  3. I think “we adopt the method proposed in [26] that uses the pixel-wise distance to attenuate the weight from the area furthest from the center of the instance” in the third paragraph of Section 3.3 of the article should be described clearly. The details of the function f in formula 2 need to be described in more detail.
  4. The article focuses on solving the problem of hard example instance segmentation and mentions the phenomenon that hard example is difficult to segment. Can you provide some pictures to explain which samples in remote sensing images are hard examples?
  5. I think that providing some updated instance segmentation algorithms to compare the loss function of the article and the basic loss function will better prove your algorithm.

Reviewer 2 Report

The authors presented an improved instance segmentation based on calculation of segmentation difficulty of each instance and shape-penalty map. The paper is well structured and the methodology seems to improve results over related work with a not too complicated, but effective upgrade.

There are some issues that should be addressed:
- It is not clear whether shape penalty mask is a new type of map, or is derived from [26].
- For some benchmarks you used a subset of a 100 images for each category. Can it be claimed that your methodology improves results in general based only on a subset? It might occur that it performs worse on a different part of a dataset, since the differences are small.
- Explanation of some acronyms is missing (e. g. VHR) 
- The term "ours" is used all over the paper, including figures. I suggest to replace the term with a more appropriate term, such as "proposed". 

Reviewer 3 Report

Comments:

  1. Why is the proposed approach suitable to be used to solve the critical problem? We need a more convinced response to indicate clearly the SOTA development and how the penalty map differentiates the various edges as you mention in your method as a core part.
  2. Please explain why to use these datasets in the experimental design and mention the names of datasets in the abstract.
  3. As you mentioned you have used mask R-CNN so it’s an existence technique. What makes the proposed method New and suitable for this unique task? What new development to the proposed method have the authors added (compared to the existing approaches)? These points should be clarified.
  4. The complexity of the proposed model and the model parameter uncertainty is not mentioned. It should be mentioned and experimentally proven to easily show the model significance.
  5. The abstract of the paper should briefly explain the summary of this work. In the said paper authors should add numerical improvements in the abstract against the SOTA.
  6. The current challenges are not crystal clearly mentioned in the introduction section.
  7. The contributions in the current version are not crystal clear. I strongly recommend to add built-wise contributions.
  8. There is no any mention of a specific research era and detailed documentation of datasets.
  9. The author needs to proper attention toward the experimental section. The performance of the proposed method should be better analyzed, commented and visualized in the experimental section.
  10. More recently-published papers in the field of deep learning should be discussed in the Introduction/literature. The authors may be benefited by reviewing more papers such as DOI: 10.3390/math8122133.
  11. The readability and presentation of the study should be further improved. The paper suffers from language problems.

Round 2

Reviewer 3 Report

The authors successfully addressed my comments and suggestions. Good Luck!